# General relativistic stochastic thermodynamics

Tao Wang,* Yifan Cai,† Long Cui,‡ and Liu Zhao§

School of Physics, Nankai University, Tianjin 300071, China

## Abstract

Based on the recent work [1, 2], we formulate the first law and the second law of stochastic thermodynamics in the framework of general relativity. These laws are established for a charged Brownian particle moving in a heat reservoir and subjecting to an external electromagnetic field in generic stationary spacetime background, and in order to maintain general covariance, they are presented respectively in terms of the divergences of the energy current and the entropy density current. The stability of the equilibrium state is also analyzed.

**Keywords:** Stochastic thermodynamics; general covariance; first law; second law; stability

## 1 Introduction

Since the publication of Sekimoto's article [3] in the 1990s, stochastic energetics has increasingly emerged as a prominent field of interest. Especially when biological experiments served as evidence to validate the rationality of stochastic energetics, this field has become one of the most active research direction in non-equilibrium statistical physics. This theoretical framework is based on the Langevin equation and the corresponding Fokker-Planck equation (FPE), and the elegance of this theory lies in its seamless establishment of the relationship between these equations and thermodynamics [4]. Although this field has been developing for nearly thirty years, there are still many interesting issues which are worth of further exploration [5, 6].

Apart from the current hot topics such as discussing specific bounds based on this theory [7,8], a more fundamental question is whether we can discuss this theory within the framework of relativity. It is well known that all physics laws should inherently abide by the principles of relativity. However, the original Langevin equation was built on top of Newtonian mechanics. This situation calls for a relativistic covariant treatment for the stochastic motion of the Brownian particle. Recently, we established a reasonable framework for relativistic stochastic mechanics in [1, 2], in which both the Langevin and Fokker-Planck equations (FPEs) are fully general relativistic covariant and are

---

*email: taowang@mail.nankai.edu.cn

†email: caiyifan@mail.nankai.edu.cn

‡email: cuilong@mail.nankai.edu.cn

§Corresponding author, email: lzhao@nankai.edu.cn

independent on the concrete choice of spacetime background except for a requirement of stationarity. As a natural continuation of these preceding works, it seems now the right time to revisit the existing stochastic thermodynamics within the framework of general relativity.

The related problem in the context of special relativity has been discussed by Pal and Deffner [9], our work extends the relevant discussions to curved spacetime. Some unnatural assumptions are discarded in our analysis. The clarification for the connection and difference between the coordinate system and the observer in [1] provides us a solid working ground to maintain covariance throughout the formalism. Thanks to such generality, our discussion is irrelevant to any specific choice of spacetime background.

This paper is organized as follows. We review the covariant formalism of relativistic stochastic mechanics in Section 2, and the corresponding content has been extended to the case of charged Brownian particle under the influence of external Maxwell field. In Section 3 and Section 4, we discuss the first and second laws of stochastic thermodynamics. To maintain fully relativistic covariance, these laws are presented in the form of divergences of the energy and entropy density currents, and the powers of gravity and of the electromagnetic field as well as the heat transfer rate play essential roles in the first law. Section 5 is devoted to the analysis on the stability of the equilibrium state, which helps to elucidate the authentic underpinnings of the long time limitation. In Section 6, we present a brief summary and outline some open questions that still need to be answered along the line of the current research.

The convention of notations follows our previous work [1,2], and it is important to distinguish random and deterministic variables, as well as objects in different spaces. The variables with tilde are random variables, and the corresponding un-tilded symbols represent their concrete realizations. The objects in different spaces (including the $(d+1)$-dimensional spacetime $\mathcal{M}$, future mass shell bundle $\Gamma_m^+$ and the future mass shell $(\Gamma_m^+)_x$ at the event $x$) are distinguished by the font of the objects and their indices. We will provide the specific meaning of new symbols when they appear.

## 2    Elements of relativistic stochastic mechanics

Starting with the geodesic equation on the spacetime manifold $\mathcal{M}$, we built the covariant Langevin equation $\mathrm{LE}_\tau$ with evolution parameter $\tau$ (i.e. the proper time of the Brownian particle) [1] by supplementing the former with contributions from the damping, the Gaussian stochastic and the additional stochastic forces. The resulting system describes the motion of a Brownian particle in the future mass shell bundle $\Gamma_m^+$,

$$
\begin{cases}
\mathrm{d}\tilde{x}_\tau^\mu = \dfrac{\tilde{p}_\tau^\mu}{m}\mathrm{d}\tau, \\
\mathrm{d}\tilde{p}_\tau^\mu = \left[\mathcal{R}^\mu{}_{\mathfrak{a}} \circ_S \mathrm{d}\tilde{w}_\tau^{\mathfrak{a}} + \mathcal{F}_{\mathrm{add}}^\mu \mathrm{d}\tau\right] + \mathcal{K}^{\mu\nu}U_\nu \mathrm{d}\tau - \dfrac{1}{m}\Gamma^\mu{}_{\alpha\beta}\tilde{p}_\tau^\alpha \tilde{p}_\tau^\beta \mathrm{d}\tau.
\end{cases}
\tag{1}
$$

Here, $(\tilde{x}_\tau^\mu, \tilde{p}_\tau^\mu)$ represents the path of the Brownian particle in the tangent bundle $T\mathcal{M}$, or more precisely in the future mass shell bundle $\Gamma_m^+ = \bigcup_{x\in\mathcal{M}}(\Gamma_m^+)_x$, with $(\Gamma_m^+)_x$ denoting the momentum

space at the spacetime event $x$ (i.e. the mass shell). Thanks to the mass shell condition, the components of $\tilde{p}_\tau^\mu$ are not all independent. $\mathcal{R}^\mu{}_\mathfrak{a}$, $\mathcal{F}_{\text{add}}^\mu$, $\mathcal{K}^{\mu\nu}$ are respectively the amplitudes of the Gaussian stochastic force, the additional stochastic force and the friction tensor, and they are all tensors on $(\Gamma_m^+)_x$. $\mathrm{d}\tilde{w}_\tau$ is the Gaussian increment taking the proper time of Brownian particle $\tau$ as the evolution parameter. The assumption that the stationary solution of the corresponding FPE is identical to the one particle distribution function (1PDF) of the reservoir particles in the long time limit helps in determining the form of the additional stochastic force [2]

$$\mathcal{F}_{\text{add}}^\mu = \frac{\delta^{\mathfrak{ab}}}{2} \mathcal{R}^\mu{}_\mathfrak{a} \nabla_j^{(h)} \mathcal{R}^j{}_\mathfrak{b}, \tag{2}$$

where $\nabla_j^{(h)}$ is the covariant derivative on $(\Gamma_m^+)_x$. $U_\nu$ is the proper velocity of the heat reservoir and also of the reservoir comoving observer *Bob*. $\Gamma^\mu{}_{\alpha\beta}$ appearing in the last term is the Christoffel connection on $\mathcal{M}$. Notice that, in eq.(2), $\mathcal{R}^\mu{}_\mathfrak{a}$ and $\mathcal{R}^i{}_\mathfrak{a}$ are the same thing but with different coordinate bases $\left\{\frac{\partial}{\partial p^\mu}\right\}$ and $\left\{\frac{\partial}{\partial \breve{p}^i} := \frac{\partial}{\partial p^i} - \frac{p_i}{p_0}\frac{\partial}{\partial p^0}\right\}$, and all tensors on the mass shell $(\Gamma_m^+)_x$ have such different representations.

In general, the proper time $\tau$ of the Brownian particle should become a random variable with respect to an arbitrarily chosen, non-comoving observer *Alice*. Therefore, a better description for the stochastic motion of the Brownian particle should be parametrized by the proper time $t$ of Alice, rather than that of the Brownian particle. This is achieved in [1,10] through a reparametrization scheme and the corresponding Langevin equation is denoted $\text{LE}_t$. The concrete form of $\text{LE}_t$ will not be used in this work, so we omit it.

In order to establish a statistical description for the Brownian particle, the first thing to be clarified is the space of states. It is crucial to notice that the configuration space must be taken as an "equal time slice" of the spacetime manifold, and as such it is inherently observer dependent. To keep things as general as possible, we adopt Alice as the observer whose proper velocity is denoted as $Z^\mu$. The proper time $t$ of Alice is made use of in making the time slicing,

$$\mathcal{S}_t = \{x \in \mathcal{M} | t(x) = t = \text{const.}\}, \tag{3}$$

where $t(x)$ is an extension of the proper time $t$ over $\mathcal{M}$ as a scalar field. The space of states is a hypersurface $\Sigma_t$ in the mass shell bundle [2],

$$\Sigma_t := \bigcup_{x \in \mathcal{S}_t} (\Gamma_m^+)_x = \{(x, p) \in \Gamma_m^+ | x \in \mathcal{S}_t\}, \tag{4}$$

which is accompanied by the timelike hypersurface orthogonal vector field $\mathscr{Z} = Z^\mu e_\mu$, where $e_\mu$ is a part of the orthogonal basis of the mass shell bundle $\{e_\mu, \partial/\partial p^\mu\}$. As mentioned earlier, $\text{LE}_\tau$ is not the proper version of the equation that describes the stochastic motion in $\Sigma_t$ and we should replace it by $\text{LE}_t$. From the point of view of stochastic thermodynamics, the more important object is the corresponding FPE. After conducting both numerical and analytical validations [1,2], it has been proved that the physical distribution $\varphi$ of the Brownian particle as observed by Alice is encoded in the solution of the reduced FPE,

$$\frac{1}{m}\mathscr{L}(\varphi) = \nabla_i^{(h)} \mathcal{I}^i[\varphi], \tag{5}$$

where

$$\mathscr{L} = p^\mu \frac{\partial}{\partial x^\mu} - \Gamma^\mu{}_{\alpha\beta} p^\alpha p^\beta \frac{\partial}{\partial p^\mu}$$

is the Liouville vector field which is also the Hamiltonian vector field associated with the Hamiltonian $H = \frac{1}{2m} g_{\mu\nu} p^\mu p^\nu$ of a free relativistic particle,

$$\mathcal{I}^i[\varphi] = \frac{1}{2} \mathcal{D}^{ij} \nabla_j^{(h)} \varphi - \mathcal{K}^{i\nu} U_\nu \varphi, \tag{6}$$

and $\mathcal{D}^{ij} = \delta^{\mathfrak{ab}} \mathcal{R}^i{}_\mathfrak{a} \mathcal{R}^j{}_\mathfrak{b}$ is the diffusion tensor in the momentum space. The vector field $\mathcal{I}^i[\varphi]$ is tangent to the mass shell, thus it can carry either a Latin index or a Greek one [2], which correspond to components under different bases. Moreover, this vector field is closely connected to the heat transfer rate from the reservoir to the Brownian particle,

$$Q[\varphi] := \int_{(\Gamma_m^+)_x} \eta_{(\Gamma_m^+)_x} Z_\nu \mathcal{I}^\nu[\varphi]. \tag{7}$$

When $\mathcal{I}^i[\varphi]$ vanishes, the heat transfer rate also vanishes.

Some characteristic quantities of the system including particle number current and energy-momentum-stress tensor can be constructed using this distribution,

$$N^\mu[\varphi] = \int \eta_{(\Gamma_m^+)_x} \frac{p^\mu}{m} \varphi N := N n^\mu[\varphi], \qquad T^{\mu\nu}[\varphi] = \int \eta_{(\Gamma_m^+)_x} \frac{p^\mu p^\nu}{m} \varphi, \tag{8}$$

where $N$ is a constant denoting the total number of the Brownian particle and $n^\mu$ is the one-particle current. Here and below we use the notation $\eta_{(\Gamma_m^+)_x}$ to denote the invariant volume element on $(\Gamma_m^+)_x$. Similar notations such as $\eta_\mathcal{M}$, $\eta_{\mathcal{S}_t}$, $\eta_{\Sigma_t}$ *etc* are all invariant volume elements on the relevant manifolds given in the suffices. The divergences of the above tensors are given as follows [2],

$$\nabla_\mu N^\mu[\varphi] = N \nabla_\mu n^\mu[\varphi] = 0, \qquad \nabla_\mu T^{\mu\nu}[\varphi] = -\int \eta_{(\Gamma_m^+)_x} \mathcal{I}^\nu[\varphi]. \tag{9}$$

In the rest of this section, we will extend the basic equations of stochastic mechanics in curved spacetime to the case of charged Brownian particle subjecting to external electromagnetic field $A_\mu$. Let $F_{\mu\nu} = \partial_\mu A_\nu - \partial_\nu A_\mu$ be the corresponding field strength, then the extension of $\mathrm{LE}_\tau$ should read

$$\begin{cases} \mathrm{d}\tilde{x}_\tau^\mu = \dfrac{\tilde{p}_\tau^\mu}{m} \mathrm{d}\tau, \\ \mathrm{d}\tilde{p}_\tau^\mu = \left[ \mathcal{R}^\mu{}_\mathfrak{a} \circ_S \mathrm{d}\tilde{w}_\tau^\mathfrak{a} + \mathcal{F}_{\mathrm{add}}^\mu \mathrm{d}\tau \right] + \mathcal{K}^{\mu\nu} U_\nu \mathrm{d}\tau - \dfrac{1}{m} \Gamma^\mu{}_{\alpha\beta} \tilde{p}_\tau^\alpha \tilde{p}_\tau^\beta \mathrm{d}\tau + \dfrac{q}{m} F^\mu{}_\nu \tilde{p}_\tau^\nu \mathrm{d}\tau. \end{cases} \tag{10}$$

The modification to the reduced FPE is fully encoded in the modified Liouville vector field [11]

$$\mathscr{L}_F = p^\mu \frac{\partial}{\partial x^\mu} + \left( q F^\mu{}_\nu p^\nu - \Gamma^\mu{}_{\alpha\beta} p^\alpha p^\beta \right) \frac{\partial}{\partial p^\mu}. \tag{11}$$

Apart from the different form of the Liouville vector field, the reduced FPE takes the same form as eq.(5), which can be verified by use of the diffusion operator approach (see the appendix in [2]),

$$\frac{1}{m} \mathscr{L}_F(\varphi) = \nabla_i^{(h)} \mathcal{I}^i[\varphi]. \tag{12}$$

In particular, the vector field $\mathcal{I}^i[\varphi]$ maintains its original form as given in eq.(6). Using the probability current of the Brownian particle defined as

$$\mathscr{J}[\varphi] := \frac{\varphi}{m}\mathscr{L}_F - \mathcal{I}[\varphi], \tag{13}$$

the reduced FPE (12) can also be written as a current conservation equation

$$\hat{\nabla}_A^{(\hat{h})}\mathscr{J}^A[\varphi] = 0, \tag{14}$$

where $\hat{\nabla}^{(\hat{h})}$ is covariant derivative operator on the mass shell bundle.

We assume that the heat reservoir is in its intrinsic equilibrium state with the 1PDF [11–14]

$$\varphi_R = \mathrm{e}^{-\alpha_R + U_\mu p^\mu / T_B},$$

where $T_B$ is the temperature as measured by the comoving observer Bob. In the long time limit, the 1PDF for the Brownian particle should approach the same form as the equilibrium distribution of the reservoir, i.e.

$$\varphi_{\mathrm{eq}} = \mathrm{e}^{-\alpha + \beta_\mu p^\mu}, \qquad \beta_\mu = U_\mu / T_B, \tag{15}$$

and it should stop receiving heat from the reservoir, i.e. $\mathcal{I}^i[\varphi] = 0$. On such occasions, the reduced FPE reduces into the Liouville equation $\mathscr{L}_F \varphi = 0$, which leads to the following constraints over $\alpha$ and $\beta_\mu$ (the equilibrium conditions)

$$\nabla_\mu \alpha + q\beta^\nu F_{\mu\nu} = 0, \qquad \nabla_{(\mu}\beta_{\nu)} = 0. \tag{16}$$

The condition over $\beta_\mu$ remains the same as in the case of neutral Brownian particle, which implies that $\beta_\mu$ should be future directed and timelike Killing. The condition over $\alpha$ is dependent on the electromagnetic field strength, thus making a difference from the neutral case.

It is easy to check that the condition $\mathcal{I}^i[\varphi] = 0$ calls for an un-altered Einstein relation as in the case of neutral Brownian particle,

$$\mathcal{D}^{ij} = 2T_B \mathcal{K}^{ij}, \tag{17}$$

and the particle number current remains divergence-free. However, the divergence of the energy-momentum tensor needs to be modified which reflects the consequence of the action of the electromagnetic field,

$$\nabla_\mu T^{\mu\nu}[\varphi] = qF^\nu{}_\mu n^\mu[\varphi] - \int \eta_{(\Gamma_m^+)_x} \mathcal{I}^\nu[\varphi]. \tag{18}$$

## 3 First law of relativistic stochastic thermodynamics

The first law of relativistic Brownian particle in the absence of an external field has been established in our previous work [2], which is presented in the form of the divergence of the energy current rather than in the change in the energy itself as in the ordinary thermodynamics,

$$\nabla_\mu E^\mu[\varphi] = P_{\mathrm{grav}}[\varphi] + Q[\varphi], \tag{19}$$

where $Q[\varphi]$ is the heat transfer rate as given in eq.(7), and

$$P_{\text{grav}}[\varphi] = -\int \eta_{(\Gamma_m^+)_x} \frac{\varphi}{m} p^\mu p^\nu \nabla_\mu Z_\nu = -T^{\mu\nu}[\varphi]\nabla_\mu Z_\nu \tag{20}$$

is the gravitational power as measured by Alice [15]. In this section, we will build the complete first law of relativistic Brownian particle in the presence of electromagnetic field.

It should be mentioned that, in the presence of electromagnetic field, the definition of energy of the charged Brownian particle is non-unique. Such non-uniqueness arises from different choices of the proper momentum of the particle, i.e. the kinematic momentum $p^\mu$ (which is proportional to the proper velocity of the particle) and the physical momentum $P^\mu = p^\mu + qA^\mu$. Each choice gives rise to a separate definition of the energy, i.e.

$$E_p = -Z_\mu p^\mu \tag{21}$$

and [16]

$$E_P = -Z_\mu P^\mu. \tag{22}$$

$E_p$ corresponds to the kinematic energy and $E_P$ corresponds to the Hamiltonian. It will be clear that the different choices of energy leads to different forms of the first law of relativistic stochastic thermodynamics, both forms are physically correct.

Following the discussions made in Appendix A, we now introduce the energy currents associated with $E_p$ and $E_P$ as follows,

$$E_p^\mu[\varphi] = \int \eta_{(\Gamma_m^+)_x} \frac{p^\mu}{m} \varphi E_p, \qquad E_P^\mu[\varphi] = \int \eta_{(\Gamma_m^+)_x} \frac{p^\mu}{m} \varphi E_P. \tag{23}$$

These currents are related to the energy-momentum tensor via the following equations,

$$E_p^\mu[\varphi] = -Z_\nu T^{\mu\nu}[\varphi], \qquad E_P^\mu[\varphi] = -Z_\nu(T^{\mu\nu}[\varphi] + qA^\mu n^\nu[\varphi]), \tag{24}$$

where $T^{\mu\nu}[\varphi]$ and $n^\mu[\varphi]$ are given in eq.(8). Using the trick discussed in Appendix A, the divergences of the above currents are calculated to be

$$\nabla_\mu E_p^\mu[\varphi] = -\int \eta_{(\Gamma_m^+)_x} \left( \frac{\varphi}{m} p^\mu p^\nu \nabla_\mu Z_\nu + \frac{q\varphi}{m} F_{\mu\nu} p^\nu Z^\mu - \mathcal{I}^\mu[\varphi]Z_\mu \right), \tag{25}$$

and

$$\nabla_\mu E_P^\mu[\varphi] = -\int \eta_{(\Gamma_m^+)_x} \left( \frac{\varphi}{m} p^\mu p^\nu \nabla_\mu Z_\nu + \frac{q\varphi}{m} p^\mu \pounds_Z A_\mu - \mathcal{I}^\mu[\varphi]Z_\mu \right), \tag{26}$$

in which $\pounds_Z A_\mu$ is the Lie derivative of $A_\mu$ along the direction of $Z^\mu$. The first and the third terms on the right hand side of eqs.(25) and (26) can be easily recognized to be the gravitational power and the heat transfer rate respectively as presented in eqs.(20) and (6). The middle terms on the right hand side of eqs.(25) and (26) can be respectively denoted as

$$P_{\text{em}}[\varphi] = -\int \eta_{(\Gamma_m^+)_x} \frac{q\varphi}{m} F_{\mu\nu} p^\nu Z^\mu = -qF_{\mu\nu} n^\mu[\varphi]Z^\nu,$$

$$P_{\mathrm{nc}}[\varphi] = -\int \eta_{(\Gamma_m^+)_x} \frac{q\,\varphi}{m} p^\mu \pounds_Z A_\mu = -qn^\mu[\varphi]\pounds_Z A_\mu. \tag{27}$$

$P_{\mathrm{em}}[\varphi]$ is obviously the power of the electromagnetic field, while $P_{\mathrm{nc}}[\varphi]$ should be understood as the power of the non-conservative part of the electromagnetic field observed by Alice. It might be helpful to point out that $P_{\mathrm{nc}}[\varphi] \neq 0$ only when $\pounds_Z A_\mu \neq 0$, i.e. when the proper velocity of Alice is not a symmetry generator of the electromagnetic field. The difference between eq.(25) and eq.(26) lies in the fact that $E_P$ absorbs the potential energy of the particle into the total energy.

Finally, both versions of the first law can be expressed in a concise form,

$$\nabla_\mu E_p^\mu = P_{\mathrm{grav}}[\varphi] + P_{\mathrm{em}}[\varphi] + Q[\varphi], \tag{28}$$

$$\nabla_\mu E_P^\mu = P_{\mathrm{grav}}[\varphi] + P_{\mathrm{nc}}[\varphi] + Q[\varphi]. \tag{29}$$

# 4   Second law of relativistic stochastic thermodynamics

The research on macroscopic irreversibility has taken central stage in the history of thermodynamics and statistical physics. After about 150 years of development, this field has grown considerably. For example, it was found that the second law of thermodynamics can also be established in stochastic thermodynamics [3], and we will give a relativistic version of the law in this section. It is worth mentioning that the fluctuation theorem gives a fairly good explanation for the origin of macroscopic irreversibility [17–22], but the relevant analysis is relatively complex, so we leave it to a separate upcoming work.

Let us now precede by separating the probability current (13) of the Brownian particle into the time-reversible and dissipative parts as did in [23–25],

$$\mathscr{J}_{\mathrm{r}}[\varphi] = \frac{\varphi}{m}\mathscr{L}_F, \qquad \mathscr{J}_{\mathrm{d}}[\varphi] = -\mathcal{I}[\varphi]. \tag{30}$$

The reversibility of $\mathscr{J}_{\mathrm{r}}[\varphi]$ can be easily understood because $\mathscr{L}_F$ is the Hamiltonian vector field which is clearly time-reversible. The dissipative part is caused by the diffusion and friction forces which breaks the time-reversal symmetry. It will be clear shortly that the entropy production is closely related to the breaking of this symmetry.

The definition of entropy in non-equilibrium states is not as clear as the entropy in the equilibrium [26], especially in the relativistic context. We could of course imitate the form of Gibbs entropy but since we have at least three different FPEs for different distribution functions [2], and each distributions have reasonable meanings in different spaces, it seems that the definition for the entropy is not unique. Fortunately, to fulfill our aim to formulate the second law of relativistic stochastic thermodynamics, it suffices to have one definition for the entropy which is non-decreasing in the course of time. Following the discussions made in [2], it is reasonable to start with the 1PDF $\varphi$ which obeys the reduced FPE. This object is the physical distribution on the future mass shell bundle. Therefore, we introduce the entropy density on the mass shell bundle of the Brownian particle as

$$s(x,p) = -\log \varphi(x,p). \tag{31}$$

This entropy density characterizes the properties of the micro states of the Brownian particle. In stochastic thermodynamics, the entropy production comes not only from the Brownian particle but also from the heat reservoir. Therefore, the relative entropy is more suitable to portray the irreversibility of the whole system. The relative entropy density reads

$$h(x,p) = -\log \frac{\varphi(x,p)}{\varphi_{\mathrm{eq}}(x,p)} = -\log \varphi - \alpha + \beta_\mu p^\mu, \tag{32}$$

where $\varphi_{\mathrm{eq}}$ is the equilibrium distribution of the Brownian particle, hence $\alpha$ and $\beta_\nu$ satisfy the equilibrium condition (16).

What we actually need to make use of while formulating the second law of relativistic stochastic thermodynamics is not the the total entropy but rather the entropy density current on the space time. The entropy density current of the Brownian particle on the spacetime is given as

$$S^\mu[\varphi] = -\int \eta_{(\Gamma_m^+)_x} \frac{p^\mu}{m} \varphi \log \varphi, \tag{33}$$

and the relative entropy density current can be expressed as the linear combination of entropy density current, the particle number current, and the energy-momentum tensor

$$H^\mu[\varphi] = S^\mu[\varphi] - \alpha n^\mu[\varphi] + \beta_\nu T^{\mu\nu}[\varphi]. \tag{34}$$

We will shortly prove that the relative entropy production rate $\nabla_\mu H^\mu[\varphi]$ is the sum of the entropy production rate of the Brownian particle and that of the heat reservoir.

Since the heat reservoir is assumed to be in local thermodynamic equilibrium, its entropy current can be expressed as [14, 27]

$$S_R^\mu = (1 + \alpha_R)N_R^\mu - \beta_\nu T_R^{\mu\nu}. \tag{35}$$

This expression is not affected by the presence of the electromagnetic field, but the equilibrium condition of the reservoir will take a form similar to that of the Brownian particle,

$$\nabla_\mu \alpha_R + q_R \beta^\nu F_{\mu\nu} = 0, \qquad \nabla_{(\mu} \beta_{\nu)} = 0, \tag{36}$$

where $q_R$ characterizes the charge of the reservoir particle which could either be vanishing or not. We assume that the reservoir is consisted purely of classical particles with no chemical reactions, its particle number current should be conserved, $\nabla_\mu N_R^\mu = 0$. However, since the reservoir exchanges energy with the Brownian particle and also with the electromagnetic field, the divergence of the total energy-momentum tensor needs to obey

$$\nabla_\mu T_R^{\mu\nu} + \nabla_\mu T^{\mu\nu}[\varphi] = F^\nu_{\ \mu}(qn^\mu[\varphi] + q_R N_R^\mu), \tag{37}$$

which means that the non-conservation of the total energy-momentum tensor is originated solely from the external field. With the help of eqs.(35–37) and eq.(18), we finally find the covariant divergence of entropy density current of the reservoir,

$$\nabla_\mu S_R^\mu = N_R^\mu \nabla_\mu \alpha_R - \beta_\nu \nabla_\mu T_R^{\mu\nu}$$

$$= -q_R N_R^\mu \beta^\nu F_{\mu\nu} - \beta_\nu \left[ F^\nu{}_\mu (qn^\mu + q_R N_R^\mu) - \nabla_\mu T^{\mu\nu}[\varphi] \right]$$
$$= -n^\mu[\varphi]\nabla_\mu \alpha + \beta_\nu \nabla_\mu T^{\mu\nu}[\varphi], \tag{38}$$

where the equilibrium condition of reservoir particles (36) is used in the second line and the equilibrium condition of Brownian particle (16) is used in the third line. The total entropy production rate is then

$$\nabla_\mu S_{\text{tot}}^\mu = \nabla_\mu S^\mu[\varphi] + \nabla_\mu S_R^\mu$$
$$= \nabla_\mu S^\mu[\varphi] - n^\mu[\varphi]\nabla_\mu \alpha + \beta_\nu \nabla_\mu T^{\mu\nu}[\varphi] = \nabla_\mu H^\mu[\varphi], \tag{39}$$

which is the desired result advocated earlier.

Using eq.(54) in Appendix A we can evaluate the total entropy production rate to be

$$\nabla_\mu S_{\text{tot}}^\mu = \nabla_\mu H^\mu[\varphi] = \int \eta_{(\Gamma_m^+)_x} \mathscr{I}[\varphi](h)$$
$$= \int \eta_{(\Gamma_m^+)_x} \frac{\mathcal{D}^{\mu\nu}}{2\varphi} \left( \frac{\partial}{\partial p^\mu}\varphi - \beta_\mu \varphi \right) \left( \frac{\partial}{\partial p^\nu}\varphi - \beta_\nu \varphi \right), \tag{40}$$

where the Einstein relation and the vanishing boundary condition for $\varphi$ at the infinity of the mass shell have been used. Since the diffusion tensor $\mathcal{D}^{ij} = \delta^{\mathfrak{ab}}\mathcal{R}^i{}_{\mathfrak{a}}\mathcal{R}^j{}_{\mathfrak{b}}$ must be a semi-positive definite quadratic form, the total entropy production rate is always nonnegative. Therefore, we arrive at the second law of relativistic stochastic thermodynamics

$$\nabla_\mu S_{\text{tot}}^\mu \geq 0. \tag{41}$$

The entropy production is always closely related to the breaking of the time-reversal symmetry. We can divide the total entropy production rate into that of the Brownian particles and of the heat reservoir, both will be shown to be proportional to the dissipative part of the probability current.

The entropy production rate of the Brownian particle reads

$$\nabla_\mu S^\mu[\varphi] = \int \eta_{(\Gamma_m^+)_x} \mathscr{I}[\varphi](-\log \varphi)$$
$$= -\int \eta_{(\Gamma_m^+)_x} \varphi^{-1} \left( \frac{\partial \varphi}{\partial p^\mu} \right) \mathscr{J}_d^\mu[\varphi], \tag{42}$$

where the trick mentioned in Appendix A is used again, and this derivation is similar to that leads to eq.(40). Substituting eq.(18) into eq.(38), the entropy production rate of the reservoir caused by the heat exchange can be expressed as

$$\nabla_\mu S_R^\mu = -\beta_\nu \int \eta_{(\Gamma_m^+)_x} \mathcal{I}^\nu[\varphi] = \int \eta_{(\Gamma_m^+)_x} \beta_\mu \mathscr{J}_d^\mu[\varphi]. \tag{43}$$

Similar result was also mentioned in Pal and Deffner's paper [9], but now it is made fully covariant. It is obvious that the time-reversible part $\mathscr{J}_r^\mu[\varphi]$ of the probability current does not contribute to the entropy production rate. All entropy productions arise as consequences of the dissipative part $\mathscr{J}_d^\mu[\varphi]$, which is the same as in the non-relativistic case [9, 25]. It is worth noticing that eq.(43) is actually the relativistic version of Clausius' identity for the heat reservoir,

$$\nabla_\mu S_R^\mu = -\frac{Q_B[\varphi]}{T_B} = \frac{Q_R}{T_B}, \tag{44}$$

where $Q_B[\varphi]$ is the heat transfer rate from the heat reservoir to the Brownian particle from the perspective of Bob, and hence $Q_R = -Q_B[\varphi]$ is the heat transfer rate from the Brownian particle to the heat reservoir.

There are some interesting consequences from the equation of entropy production rate. Assuming that the diffusion tensor $\mathcal{D}^{ij}$ is a full-rank matrix and hence strictly positive definite, the condition for zero entropy production rate becomes

$$\left( \frac{\partial}{\partial p^\mu} - \beta_\mu \right) \varphi = 0. \tag{45}$$

This is equivalent to the thermal equilibrium condition $\mathcal{I}[\varphi] = 0$, thanks to the covariant Einstein relation (17). Therefore, the equilibrium state is the unique state which saturates the second law. However, if the diffusion tensor is not full-rank, implying that there exists some nonzero vector field $\mathcal{C}^i[\varphi]$ on the momentum space obeying $\mathcal{D}_{ij}\mathcal{C}^i[\varphi]\mathcal{C}^j[\varphi] = 0$, the condition for zero entropy production rate will become

$$\left( \frac{\partial}{\partial p^\mu} - \beta_\mu \right) \varphi = \mathcal{C}_\mu[\varphi]. \tag{46}$$

This leads to the possibility for the existence of certain state which is different from the prescribed equilibrium state but still with zero entropy production rate. This situation is not unexpected. The presence of a degenerate diffusion tensor implies the existence of certain degrees of freedom that are decoupled from the heat reservoir. The 1PDF of these states obeys the Liouville equation and the entropy remains conserved. In the next section, we will discuss how the rank of the diffusion tensor affects the ability of the Brownian particle to reach the equilibrium state.

## 5 The stability of equilibrium state

In Ref. [2], we assumed that the Brownian particle can always reach thermal equilibrium with the heat reservoir in the long time limit, without delving into the dynamical process for the Brownian particle to evolve from a non-equilibrium state to the equilibrium. To tackle this problem, the Lyapunov theorem offers a powerful tool.

For the sake of convenience, we make an equivalent statement of the Lyapunov theorem: if there exists a functional for a given system that satisfies the condition of possessing at least one maximum point, with its time derivative strictly positive except at these maximum points, then the system is deemed asymptotically stable [28]. This functional is commonly referred to as the Lyapunov functional. The maximum points of the Lyapunov functional represent the stable states of the system. In the event that a system in a stable state experiences minor disturbances, as long as these disturbances do not surpass the local minimum of the Lyapunov functional, the system will return to its original stable state. Notably, if the maximum point of the Lyapunov functional is unique, no matter which initial state the system starts from, it will always reach the stable state. Such a property is called global asymptotically stable.

In our case, the expectation value of the relative entropy

$$\bar{H}_t[\varphi] := \int_{\Sigma_t} \eta_{\Sigma_t} \frac{Z_\mu p^\mu}{m} \varphi \log \frac{\varphi}{\varphi_{\text{eq}}} \tag{47}$$

plays the role of Lyapunov functional of the Brownian particle. We now try to impose a small perturbation to the distribution function and calculate the variation of the above functional. The effect of the perturbation is to drive the system from one distribution to another, so the variation only acts on the distribution function. The result of the variation reads

$$
\begin{aligned}
\delta \bar{H}_t[\varphi] &= \int \eta_{\Sigma_t} \frac{Z_\mu p^\mu}{m} \delta \left[ \varphi \left( \log \varphi + \alpha - \beta_\nu p^\nu \right) \right] \\
&= \int \eta_{\Sigma_t} \frac{Z_\mu p^\mu}{m} \left[ \left( 1 + \log \varphi + \alpha - \beta_\nu p^\nu \right) \delta \varphi + \varphi^{-1} (\delta \varphi)^2 + O((\delta \varphi)^2) \right] \\
&= \int \eta_{\Sigma_t} \frac{Z_\mu p^\mu}{m} \left[ \left( \log \varphi + \alpha - \beta_\nu p^\nu \right) \delta \varphi + \varphi^{-1} (\delta \varphi)^2 + O((\delta \varphi)^2) \right],
\end{aligned}
\tag{48}
$$

where, in the third line, the particle conservation has been used. It is obvious that the first order variation is zero and the second order variation is negative at the equilibrium state, so the equilibrium state is a maximum point of $\bar{H}_t[\varphi]$. Furthermore, the second order variation is not only negative at the equilibrium state but also at any states. This implies that no critical points like the one illustrated in Fig.1 could appear in such systems, and hence the relative entropy of the Brownian particle is globally convex with only one maximum.

Using eq.(56) and the second law, the time derivative of $\bar{H}_t[\varphi]$ can be confirmed to be non-negative

$$\frac{\mathrm{d}}{\mathrm{d}t} \bar{H}_t[\varphi] = \int_{\mathcal{S}_t} \eta_{\mathcal{S}_t} \nabla_\mu H^\mu[\varphi] |\nabla t|^{-1} \geq 0. \tag{49}$$

However, the Lyapunov theorem requires it to be strictly positive except at the equilibrium state. According to the discussion made in the end of the last section, if the diffusion tensor is full-rank, the above inequality can only be saturated in the equilibrium state. Therefore, $\bar{H}_t[\varphi]$ is a Lyapunov functional of the Brownian particle. Fig.1 illustrates the general behavior of a Lyapunov functional with the inclusion of an equilibrium state and an artificial metastable state. In such cases, it is possible to drive the system away from the equilibrium state into the metastable state or vice versa by appropriate perturbations. Now, assuming a full-rank diffusion tensor, the above possibility is excluded because of the global asymptotic stability. Therefore, we can draw the conclusion that the relativistic charged Brownian particle system exhibits a tendency to evolve towards an increase in relative entropy, and the eventual state of evolution is characterized by the unique equilibrium distribution.

However, if the diffusion tensor is not full-rank, there will be some states other than the equilibrium state with zero entropy production rate. The Lyapunov functional may stop growing when the Brownian particle enters such states, and the distribution function no longer evolves toward the equilibrium one. Therefore, in the situation that the diffusion tensor is not full-rank, the Brownian particle may not be able to reach the equilibrium state.

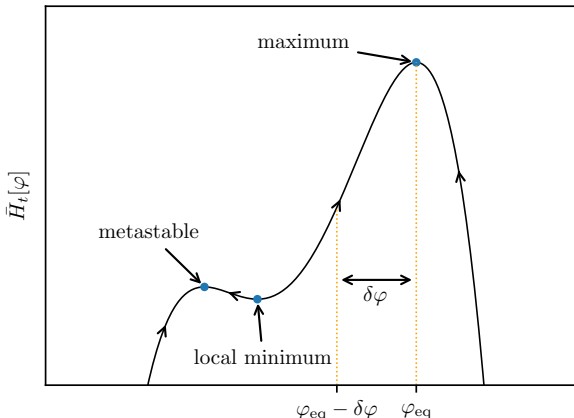

Figure 1: The curve for an artificial Lyapunov functional. In the case with a full-rank diffusion tensor, our discussion rules out the possibility for the Lyapunov functional of the charged relativistic Brownian particle to exhibit metastable and critical (i.e. inflection) points.

# 6 Concluding remarks

With the aid of covariant formalism of relativistic stochastic mechanics, the first law and the second law of general relativistic stochastic thermodynamics have been established for a system of charged Brownian particle subjecting to an external electromagnetic field. The fully covariant formulation has assisted in elucidating numerous conceptual issues. The final discussion about the stability of the equilibrium distribution helps us to clarify that the equilibrium state of Brownian particle is the state with maximum entropy.

As a potential application scenario of the generic construction, we expect that a charged Brownian particle moving in Reissner-Nordström spacetime may be an excellent example for which the theory is applicable. In doing so, the relationship between non-equilibrium statistical physics in curved spacetime and black hole thermodynamics might become an interesting subject of future study.

On the other hand, we have discussed only the first and second laws in this work, but a deeper understanding about the origin of the irreversibility in curved spacetime still calls for further investigation. At present, the best approach for the origin of irreversibility is based on various fluctuation theorems [17–22]. However, fluctuation theorems have not yet been realized in the framework of general relativity. We hope to come back to this subject in some upcoming works.

# A    Currents on spacetime and their divergences

Let $\mathscr{O}(x, p)$ be a scalar field on the mass shell bundle, then the expectation value of $\mathscr{O}$ observed by Alice reads

$$\bar{\mathscr{O}}_t[\varphi] := -\int \eta_{\Sigma_t} \mathscr{Z}_A \mathscr{J}^A[\varphi] \mathscr{O} = -\int \eta_{\mathcal{S}_t} Z_\mu \int \eta_{(\Gamma_m^+)_x} \frac{p^\mu}{m} \varphi \mathscr{O}. \tag{50}$$

Since $\mathscr{Z} = Z^\mu e_\mu$ is the unit normal vector of $\Sigma_t$ and $\mathscr{J}[\varphi]$ is the probability current, $-\mathscr{Z}_A \mathscr{J}^A$ is clearly the probability density on $\Sigma_t$. The momentum space integral in eq.(50) is simply a current on spacetime manifold,

$$\mathscr{O}^\mu[\varphi] := \int \eta_{(\Gamma_m^+)_x} \frac{p^\mu}{m} \varphi \mathscr{O}. \tag{51}$$

Therefore, the expectation value can also be written as a surface integral of $\mathscr{O}^\mu[\varphi]$ on the configuration space,

$$\bar{\mathscr{O}}_t[\varphi] = -\int_{\mathcal{S}_t} \eta_{\mathcal{S}_t} Z_\mu \mathscr{O}^\mu[\varphi], \tag{52}$$

which implies that $\mathscr{O}^\mu[\varphi]$ is the density current of $\mathscr{O}$.

Let $V$ be an arbitrary region in the spacetime manifold $\mathcal{M}$, and $\Gamma = \{(x, p) \in \Gamma_m^+ | x \in V\}$ is the corresponding region in the mass shell bundle. Let $y^\mu$ be the unit normal vector of the boundary $\partial V$, the corresponding unit normal vector of $\partial\Gamma$ is then $\mathscr{Y} = y^\mu e_\mu$. By the Gauss theorem, we have

$$\int_V \eta_{\mathcal{M}} \nabla_\mu \mathscr{O}^\mu[\varphi] = \int_{\partial V} \eta_{\partial V} y_\mu \mathscr{O}^\mu[\varphi] = \int_{\partial\Gamma} \eta_{\partial\Gamma} \frac{y_\mu p^\mu}{m} \varphi \mathscr{O} = \int_{\partial\Gamma} \eta_{\partial\Gamma} \mathscr{Y}_A \mathscr{J}^A[\varphi] \mathscr{O}$$

$$= \int_\Gamma \eta_{\Gamma_m^+} \mathscr{J}^A[\varphi] \hat{\nabla}_A^{(\hat{h})} \mathscr{O} = \int_V \eta_{\mathcal{M}} \int_{(\Gamma_m^+)_x} \eta_{(\Gamma_m^+)_x} \mathscr{J}[\varphi](\mathscr{O}). \tag{53}$$

Since the region $V$ is arbitrary, the integrand on the right hand side and the left hand side should be equal to each other. Hence, the divergence of $\mathscr{O}^\mu[\varphi]$ is

$$\nabla_\mu \mathscr{O}^\mu[\varphi] = \int_{(\Gamma_m^+)_x} \eta_{(\Gamma_m^+)_x} \mathscr{J}[\varphi](\mathscr{O}). \tag{54}$$

In relativistic thermodynamic relation, the time derivative of a physical quantity is usually replaced by the divergence of the associated current. This can be achieved as follows. Let $\mathcal{S}_{t_1}$ and $\mathcal{S}_{t_2}$ be two Cauchy surfaces and $V$ be the region enclosed by a cylindrical surface with $\mathcal{S}_{t_1}$ and $\mathcal{S}_{t_2}$ playing as the upper and bottom. Then, integrating eq.(54) over $V$, we get, by Gauss theorem and co-area formula [29, 30], the following equation,

$$\bar{\mathscr{O}}_{t_2}[\varphi] - \bar{\mathscr{O}}_{t_1}[\varphi] = \int_V \eta_{\mathcal{M}} \nabla_\mu \mathscr{O}^\mu[\varphi] = \int_{t_1}^{t_2} \mathrm{d}t \int_{\mathcal{S}_t} \eta_{\mathcal{S}_t} \nabla_\mu \mathscr{O}^\mu[\varphi] |\nabla t|^{-1}. \tag{55}$$

Therefore, the time derivative of the expectation value is the integral of the divergence of the current times $|\nabla t|^{-1}$ on the configuration space,

$$\frac{\mathrm{d}}{\mathrm{d}t} \bar{\mathscr{O}}_t[\varphi] = \int_{\mathcal{S}_t} \eta_{\mathcal{S}_t} \nabla_\mu \mathscr{O}^\mu[\varphi] |\nabla t|^{-1}. \tag{56}$$

# Acknowledgement

This work is supported by the National Natural Science Foundation of China under the grant No. 12275138.

# Data Availability Statement

This research has no associated data.

# Declaration of competing interest

The authors declare no competing interest.

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
