# Peer review of "General relativistic stochastic thermodynamics"

_SciPost Physics_

## Round 1 · Referee Report · Anonymous (Referee 1) · 2024-9-9

Strengths

  1. These laws established for a charged Brownian particle are new.
  2. The analysis of stability of the equilibrium state sounds good.

Report

In this paper, the authors established the first law and the second law of general relativistic stochastic thermodynamics for a system of charged Brownian particle subjecting to an external electromagnetic field. The results are very meaningful and interesting for further understanding the relativistic stochastic thermodynamics. Further discussion on the following aspects would also benefit this research. 1. The introduction of Eq.(1) in the text is somewhat sudden, and some brief explanation should be provided. 2. What is the fundamental difference between the results in the article and the phase transition in reference 7? 3. What physical significance can be reflected by the essential difference between the first and second laws obtained in the text and these in equilibrium thermodynamics? 4. Figure 1 shows that the charged relativistic Brownian particle exhibit critical behavior. Is there any further discussion on the critical exponent?

After these minor revisions, I would like to recommend accepting this paper.

Requested changes

  1. The introduction of Eq.(1) in the text is somewhat sudden, and some brief explanation should be provided.
  2. What is the fundamental difference between the results in the article and the phase transition in reference 7?
  3. What physical significance can be reflected by the essential difference between the first and second laws obtained in the text and these in equilibrium thermodynamics?
  4. Figure 1 shows that the charged relativistic Brownian particle exhibit critical behavior. Is there any further discussion on the critical exponent?

Recommendation

Ask for minor revision

  • validity: top
  • significance: high
  • originality: high
  • clarity: high
  • formatting: excellent
  • grammar: excellent

Author:  Liu Zhao  on 2024-10-04  [id 4835]

(in reply to Report 1 on 2024-09-09)

We thank you for the valuable comments according to which we have made some modifications to the manuscript.

  1. The introduction of Eq.(1) in the text is somewhat sudden, and some brief explanation should be provided.

Reply: This has been fixed at the beginning of Section 2.

  1. What is the fundamental difference between the results in the article and the phase transition in reference 7?

Reply: Presumably what you mentioned as reference 7 were reference 9 in the original manuscript which now becomes reference 11. However, neither that reference nor our manuscript deals with phase transition. What we did in section 5 is to argue that provided the diffusion tensor is full-rank, the equilibrium state is guaranteed to be stable, which implies that the meta-stable point in the hypothetical curve depicted in Fig.1 does not exist.

  1. What physical significance can be reflected by the essential difference between the first and second laws obtained in the text and these in equilibrium thermodynamics?

Reply: The essential differences between the two laws in stochastic thermodynamics and those in standard thermodynamics are explained in the two novel paragraphs in the beginning of the first section and in the first paragraph in section 6. Such differences make it possible to describe the behavior of small systems with only a few degrees of freedom in terms of stochastic thermodynamics, just like how normal macroscopic systems are described in terms of standard thermodynamics.

  1. Figure 1 shows that the charged relativistic Brownian particle exhibit critical behavior. Is there any further discussion on the critical exponent?

Reply: There is no criticality in our analysis and hence no critical exponents in our study.

---

## Round 1 · Referee Report · Anonymous (Referee 2) · 2024-9-25

Strengths

The paper aims to formulate the first and second laws of stochastic thermodynamics in the framework of general relativity. This is a useful contribution.

Weaknesses

The main problem with the paper is that it is not particularly pedagogical, assuming on the part of the reader much knowledge of the stochastic formalism. In particular it lacks:

1) A brief review of non-relativistic stochastic thermodynamics, to set the work in context

2) Clear definitions of some of the quantities given in the paper. For example, the definition of current in (13,14) is unclear re indices: $L_F$ appears to be a scalar, but $I[\phi]$ is a vector. In (15), are $\alpha, \beta$ functions of x? What is $N^\mu$ in (35)?

Report

I would say that the Journal's acceptance criteria are not yet met. If the authors wish to open a new pathway in an existing or a new research direction, they need to make the paper more accessible to a broader audience.

Recommendation

Ask for major revision

  • validity: good
  • significance: good
  • originality: high
  • clarity: low
  • formatting: good
  • grammar: good

Author:  Liu Zhao  on 2024-10-04  [id 4836]

(in reply to Report 2 on 2024-09-25)

We greatly appreciate your comments regarding the non-pedagogical aspects of our manuscript. According to these comments, we have made substantial modifications in order to make it more self-contained.

1) We have introduced two novel paragraphs in the beginning of section 1 which present a brief introduction for stochastic thermodynamics. This also incorporates two new references [5], [6] which are important landmarks of stochastic thermodynamics.

2) To clarify the notations and conventions, we also made various modifications and explanations. These appear, e.g. on page 4, round eq.(7)-(8); pages 5-6, after the equation that follows eq. (15) and until eq. (18); page 9, right after eq.(37), etc.

We hope these modifications are sufficient to make the paper more self-contained.

---

## Editorial Decision

resubmitted